# Using GIS to Explore the Potential of Business Rating Data to Analyse Stock and Value Change for Land Administration: A Case Study of York

**Paul Greenhalgh** [1,*]**, Helen King** [2]**, Kevin Muldoon-Smith** [1]**, Adejimi Adebayo** [3] **and Josephine Ellis** [1]

1   Department of Architecture and Built Environment, Faculty of Engineering and Environment, Northumbria University, Newcastle, Tyne NE1 8ST, UK; k.muldoon-smith@northumbria.ac.uk (K.M.-S.); josephine.ellis@northumbria.ac.uk (J.E.)
2   Department of Geography and Environmental Sciences, Faculty of Engineering and Environment, Northumbria University, Newcastle, Tyne NE1 8ST, UK; helen.m.king@northumbria.ac.uk
3   Department of Property Management and Development, School of Architecture, Design and the Built Environment, Nottingham Trent University, 50 Shakespeare Street, Nottingham NG1 4FQ, UK; adejimi.adebayo@ntu.ac.uk
*   Correspondence: paul.greenhalgh@northumbria.ac.uk; Tel.: +44-(0)191-2274593

**Abstract:** This study explores the potential of GIS to map and analyse the distribution, stock and value of commercial and industrial property using rating data compiled for the purposes of charging business rates taxation on all non-residential property in the UK. Rating data from 2010, 2017 and 2019, comprising over 6000 property units in the City of York, were filtered and classified by retail, office and industrial use, before geocoding by post code. Nominal rateable values and floor areas for all premises were aggregated in 100 m diameter hexagonal grid and average rateable value calculated to reveal changes in the distribution and value of all employment floorspace in the City over the last decade. Temporospatial analysis revealed polarisation of York's retail property market between the historic city centre and out-of-town locations. Segmenting traditional retail from food and drink premises revealed growth in the latter has mitigated the hollowing out of the city core. This study is significant in developing a replicable and efficient method of using GIS, using a nationally available rating dataset, to represent changes in the quantum, spatial distribution and relative value of employment floorspace over time to inform local and national land administration, spatial planning and economic development policy making.

**Keywords:** business rates; taxation; valuation; commercial property; England and Wales; retail; office; industrial; employment land; cadastre

---

## 1. Introduction

One of the challenges for local government officers, surveyors, property managers and administrators is the generation of meaningful and timely information and analysis to inform strategic management of the public sector estate and strategic policy development across the wider built environment. In today's world of big and open data, the challenge is less the generation of urban property data, but more how this information can be exploited and integrated successfully into contemporary public sector strategic property management and wider decision making. Wyatt [1] identified a large number of applications of GIS in land and property management, including: property conveyancing; local authority property management (council tax and rates, maintenance, identification of under-used property); development planning; agriculture and habitat management. Central to this

is the opportunity afforded by 'big data' for society and its institutions to glean insight into urban land and property markets from comprehensive datasets that are increasingly available open source or under licence in digital form. However, Jozefowicz et al. [2] found that the exploitation of geospatial data in the UK was still hampered by a number of factors, including: lack of awareness of the potential of the data; data ownership and availability issues; shortage of skills; data quality and consistency.

One such big dataset in the realm of land administration and taxation is the National Non-Domestic Rating list that is used in the UK to levy business rates on the occupiers of all non-domestic (non-residential) property. Business rates income is an increasingly important component of the UK's national tax regime, with the Ministry of Housing, Communities and Local Government predicting that it will raise £25 bn in 2019–2020, an increase in £206 m from the previous year [3]. Local authorities in England and Wales are increasingly reliant on income from business rates due to £16 bn reduction in central 'block grant' funding since 2010 [4]. Since 2013 all local authorities in England and Wales retain 50% of their business rates, with pilot authorities retaining 100% of their business rates income since 2017 [5]. This article explores the potential for central and local government and its agencies to exploit this comprehensive national dataset to explore the distribution of commercial property, its quantity and value by adding geospatial and temporal dimensions in visualising the data through the application of Geographical Information Systems (GIS)

The ability to analyse metadata pertaining to geolocated points within a GIS provides researchers and policy makers with the ability to investigate spatial and temporal patterns in data and is a fundamental methodological approach in a diverse range of disciplines including epidemiology [6,7], criminology [8] and political science [9]. In the discipline of real estate economics, the distribution of the quantity and price of both commercial and residential property can be evaluated. In many cases, this leads on to an investigation of the relationship between property price and one or more spatial variables. For example, Seo et al. [10] investigated commercial property values in relation to light rail and highway infrastructure, considering both their negative and positive effects. Koramaz and Dolmeci [11] explored the spatial determinants of house prices in Istanbul whilst Atreya et al. [12] specifically investigated "implicit flood risk premium", revealing a discount in property prices in previously-inundated areas than in comparable areas that had not flooded. Cellmer and Trojanek [7,13] used data on apartment sales to create a price map with which to explore the influence of positive and negative externalities such as proximity to green spaces or "bad neighbours" as sources of noise or pollution; they also mapped temporal variability—the distribution of the average annual price change. Netek et al. [8,14] demonstrated that value maps can be used to better understand the locational preference of creative industries. Despite such examples, there has been relatively little progress made in exploiting Rating List data in the UK since the pioneering work of Thurstain-Goodwin and Unwin in 2000 [15], the Office of the Deputy Prime Minister (ODPM) and Centre for Advanced Spatial Analysis (CASA) in 2002 [16], and Katyoka and Wyatt in 2008 [17]. Indeed, the UK Government's own Cabinet Office recently admitted that whilst it recognises that better use of geospatial data in the public sector would create additional economic and social value, this has yet to be accurately estimated [18].

Geospatial analysis of commercial property data also presents us with an opportunity to investigate whether commercial and industrial location theories, established by the likes of Weber [19], Losch [20], Isard [21] and Alonso [22], in the early to mid-twentieth century, are still relevant in this age of private mobility, footloose industries and ICT-enabled e-commerce and business. In particular, retail land use theories such as Central Place [23], Spatial Interaction [24], Bid Rent [25] and Minimum Differentiation [26], are being challenged as never before, by e-tailing and next-day fulfilment, as consumers continue to move from 'bricks to clicks'. At the same time, researchers should be mindful that causal relationships are unpredictable; relationships are interactive, dynamic and complex: social, environmental and economic factors affect one another, including proximity to resources and to markets, transport infrastructure, suitability of accommodation and topography, agglomeration economics and state intervention [27,28]

In the UK, all non-residential property is taxed on the basis of its 'rateable value' (RV), which is benchmarked, or revalued, by the Valuation Office Agency (hereafter referred to as the VOA), on behalf of Government, every 5–7 years [29]. Due to the predominantly privately owned, complex and fragmented tenure of commercial real estate in the UK, RVs are calculated for hereditaments, which represent smaller units of property rather than whole buildings. A hereditament is defined by Myers and Wyatt [30] as:

> *'a piece of real, inheritable and taxable property on which (business) rates can be charged. A hereditament generally corresponds to an extent of floor space suitable for a single occupant and might comprise a piece of land, a number of separate buildings, a single complete building, one or more floors within a building, or part of one floor.'* (Myers and Wyatt (2007), p. 288)

According to the VOA [31] there are approximately 1.95 million hereditaments in England and Wales (Scotland and Northern Ireland operate separate systems based on broadly similar principles). Of those hereditaments where floorspace data are published, there are approximately 7 billion sq ft (659 million sq m) of floorspace, 60% of which is industrial, 13% offices, 16% retail and 11% other. Local authorities in England and Wales bill occupiers and owners of all non-domestic property for payment of rates using the National Summary Valuation Data Set, which is created and maintained by the VOA on behalf of the UK Government. The non-domestic rating dataset is published online in order to allow parties with an interest in a property to check rating valuations and that of comparable properties, but also to enable anyone to exercise their right to view rating assessments in the compiled rating lists [32]

The VOA generates and holds information on every single non-residential premise in England and Wales, along with postcodes, RV, business type and total floor space, the list of which is provided to all local authorities as the basis on which they levy business rates. These are calculated by multiplying the RV (based on the market rent at the date of valuation) by the business rates multiplier set by central government (the standard rate is currently just over 50p in the £ or 50%) [33]. Periodic downward adjustment to the national business rate multipliers coincides with business rate revaluation and the introduction of the 'new' rating list which occurred in 2000, 2005, 2010 and 2017.

All non-domestic premises are revalued periodically based on an antecedent date of valuation. Thus, the most recent business rates revaluation, 1st April 2017, based on each property's RV on 1st April 2015 [29] effectively means that RVs are already two years out of date when the 'new' rating list is introduced. Similarly, the previous 2010 rating list was based on 2008 values and the next revaluation, due in 2021, is based on market values as of at 1st April 2019. It is worth noting that the date of valuation of the 2010 rating list (1st April 2008) coincided with the pre-recession peak of the property market in the UK, resulting in business rates being higher for many premises than 'market' rents for seven years [4]. In 2017, the UK Government announced that revaluations would take place every three years instead of five years after the next revaluation in 2022. The Government subsequently announced that the next revaluation was to be brought forward to 2021, with the following revaluation taking place in April 2024 [34].

It is pertinent to note that, unusual among developed economies, the UK does not have a single, complete cadastre [35]. Even the Land Registry, which records the details of all property transactions in the country, does not have records pertaining to the entire country, since registration has only been compulsory since 1990 and some properties have not changed hands in many decades or even centuries; nor does it record the exact boundaries of parcels of land [36,37]. Effectively, the UK has two partial cadastres, the Rural Land Register, a database of agricultural land with sizes and geo-references, linked to farmers and their applications for EU funding; and the fiscal cadastre consisting of the VOA's database, together with council tax data [35].

As Dale and McClaren [38] observe, whilst the need for land (and property) information in support of development becomes ever more urgent, there is opportunity to improve land administration systems, driven by developments in technology, and land and property datasets that grow ever larger. Thus, the VOA's comprehensive dataset affords the opportunity not only to represent the quantity ($m^2$)

and relative value of non-domestic property in the UK at the date of the new rating list being published (albeit with a 2 year time-lag), but also to capture changes in the quantum of floorspace and value between census points.

The research seeks to explore the potential to use the VOA's dataset to create a methodology that can be used to portray, analyse and manage the non-domestic real estate at a local level, as the backdrop against which robust strategic decisions about capital investment and spatial interventions can be made. Such analysis would potentially be of use to metropolitan and city mayors, chief executives of local authorities, public sector estate managers, spatial and transport planners, economic development officers, town and city centre managers and other functions that require comprehensive representation of changes in the stock and performance of all commercial and industrial within their jurisdiction, at a variety of spatial scales.

Specifically, all local authorities in the UK need to conduct regular employment land reviews. These are currently subsumed within Housing and Economic Land Availability Assessments (HELAA) that require them to capture the quantum and distribution of all employment floorspace in their area by bulk class, relative values (strength of market demand) and performance between census points (periodic rating revaluations) to reveal changes in stock and value (tell tales of market changes). The aims of such reviews are to determine whether existing employment land and buildings meet the needs of the economy and identify how much additional land should be allocated for employment use within a local authority's jurisdiction [39]. Typically, local authorities pay private sector consultants to carry out such reviews on their behalf when they already have a comprehensive dataset of all employment floorspace in their borough or district. It is surprising that local authorities do not make more use of the VOA rating list as the basis of such studies, as it represents, by bulk class or subsector, the distribution and quantum of all employment floorspace in any given local council district.

Thus, the aim of this research project is to explore the potential of non-domestic business rates (big) dataset to capture/represent distribution and change in stock and value of commercial and industrial property (hereditaments) in England and seek to answer the following questions for a given area:

1.  How has the stock of retail, office and industrial floorspace changed between 2010 and 2017?
2.  What is the efficacy of using geospatial techniques and rating list data to measure stock and value change?
3.  What is the most effective way of visualising business rates data in GIS to analyse the quantum and spatial distribution of employment floorspace?

## 2. Materials and Methods

### 2.1. General Approach

This research seeks to develop a general methodology for analysing and portraying the spatial distribution and temporal performance of commercial real estate markets using a three stage approach (Figure 1), comprising the three domains of data classification, quantification, spatial granularity, connected by two interfaces.

| Data classification | INTERFACE 1 | Data quantification | INTERFACE 2 | Spatial granularity |
|---|---|---|---|---|
| Typology | | | | |
| Criteria | | Measurement Scale | | Resolution |
| Segmentation | | Measurement Unit | | Precision |
| Filtering | | Calibration | | Positioning |
| Thresholds | | | | |

(Author's own)

**Figure 1.** Three Stage Approach to spatial analysis of commercial real estate markets.

*2.2. Methodology*

Commercial and industrial property data, in the form of the 'National Non-Domestic Rating' (NNDR) list, were supplied in an Excel spreadsheet. Each record pertains to a single hereditament, containing the name and address of each business, the business' type of activity, the floorspace of the premises, and its RV. The total number of hereditaments in York was 5953 in 2010, 6562 in 2017 and 6802 in 2019. This increase in the number of hereditaments between revaluations is consistent with national trend where the number of hereditaments increases over time but not necessarily aggregate floorspace.

Filtering and classification of data into three bulk classes followed the approach developed by Muldoon-Smith et al. [40]. Each record was classified by the VOA as falling within one of several hundred Special Categories (SCats) denoting the type of business. These are grouped into four aggregate categories—the three "bulk classes" of retail, office and industrial properties, and a catch-all "other" category which includes a wide variety of properties (i.e., leisure, education and sui generis uses; easements, wayleaves, advertising hoardings, bus shelters, sponsored roundabouts, telecommunications infrastructure etc.). Each bulk class contained a sufficiently large subset of records, pertaining to hereditaments with some common features, for broad trends to be observed and outliers to be identified. For example, in York in 2019, after filtering and classification, the dataset comprised 1210 industrial, 1522 office and 1867 retail property units, respectively.

Floorspace data were obtained by linking the NNDR data to a second VOA dataset, known as the 'Summary valuations list'. This dataset provides more detailed information as to how the RV for each hereditament was calculated, which, for most types of commercial property, is based on a calculation of the open market rental value per $m^2$ of a given type of property in a given area. Hereditaments were geocoded by postcode using the online geocoding tool provided by Bell [41]. Three different datasets were generated in this way. One was based on records from the 2010 revaluation of commercial property, another was based on the 2017 revaluation and the final one was based on the complete rating list as at April 2019.

RV and floorspace ($m^2$) for hereditaments within each bulk class were then aggregated within a 100m diameter hexagonal grid and RV per metre squared ($RV/m^2$) was calculated for each grid tile. The use of a hexagonal grid was determined by a process of experimentation with different geographies. As Ben-Joseph and Gordon [42] note, the advantage of hexagonal or octagonal street layouts in the built environment is that they permit movement in six or eight directions, whereas square grids do not permit diagonal movement. A related point with the use of hexagonal grids in spatial analysis is that a grid tile's adjacent neighbours in all directions are equidistant, forming a ring around it; this makes it easier to analyse movement or connectivity as exemplified by Burdziej [43]. In our study, the use of hexagons helped to address the modifiable area unit problem (MAUP) first identified by Openshaw and Taylor [44]: because streets in York generally run diagonally relative to the north axis, the hexagonal grid offered the potential for closer alignment with the underlying street pattern and therefore, reduced boundary problems. It was determined that the use of a coarser-grained grid (500m diameter) was insufficiently sensitive to display patterns of clustering, growth and decline across a given urban area. The use of smaller hexagons, meanwhile, was too fine-grained for general trends to be observed.

The process was then repeated using a different geography—the polygons indicating the coverage of each postcode within the study area (i.e., in theory, the polygon containing all the addresses that are classified within a given postcode). The aggregated results could then be displayed in choropleth maps to indicate patterns of growth and decline across the study area, and within the city centre. It is worth noting that in map form, the impression of variability in $RV/m^2$ depends on appropriate classification, since, if the same categories are employed as for retail properties, most values fall into the lowest-value category in which properties have a $RV/m^2$ of less than £100/$m^2$. In all cases, classification was by the Pretty Breaks function in QGIS, modified by conflating higher-valued classes and by altering classification breaks to make up for the distorting effect of outliers.

*2.3. Methodological Caveats*

2.3.1. Geocoding

Items were geocoded by postcode which meant that rather than being geocoded to the exact location of the relevant hereditament, they were actually geocoded to the centroid of the relevant postcode polygon. As Jacquez notes, the fundamental process of converting text-based addresses into geographic coordinates is often prone to positional error and the difference between a true location and that returned from the geocoded address is not routinely addressed [45]. Postcodes generally contain a similar number of addresses; postcode polygons vary in size according to the density of development and addresses, within the area in question. Therefore, within densely populated areas, the granularity of the resultant mapping is greater when postcode polygons are used, because they cover a smaller area than grid hexagons. They also tend to correspond well to the underlying street pattern because they contain a certain group of addresses in close proximity. However, in less-densely-populated areas, postcode polygons cover larger areas, often covering relatively large areas of undeveloped or unoccupied land. This leads to some less than informative, perhaps misleading, results where, for example, a large amount of commercial property is found on a site that falls within a very large postcode polygon; following aggregation, it is not possible to tell which part of the larger polygon is commercially attractive, and which is not. This is particularly true for industrial property, which is often located in areas with large postcode areas. If values are aggregated within any other polygon geography, not based on postcode polygons, there is the potential for values being aggregated within the wrong polygon altogether (i.e., the correct location should be within polygon A, but the centroid of the relevant postcode polygon falls within polygon B).

2.3.2. Inertia in the Stock

There is likely to be a time-lag affecting the relationship between the indicator, and market demand. The quantity (m$^2$) of commercial property changes only when units are demolished, constructed, or when existing properties undergo a change of use. This may take place some time after the property falls out of its original use. Consequently, for example, the former Terry's factory in York, which closed in 2005, was still recorded within the 2010 dataset.

2.3.3. Absence of Vacancy Data

There is one significant shortcoming in the VOA's data—they do not indicate whether the property is occupied. When businesses fail, in the first instance, it is likely to lead to an increase in vacant properties rather than a decrease in properties per se. From the point of view of local authorities' immediate finances, this may not be very important, since landlords are still liable to pay rates on empty properties, after a 3-month reprieve [46]. In order for policy makers to understand the relative performance, resilience or vulnerability of employment locations, the levels of occupation and vacancy of business units needs to be captured and portrayed.

2.3.4. Inertia in Valuation

All hereditaments in the 2010 and 2017 datasets are ascribed RVs at the respective antecedent date of valuation, that are two years out of date at publication of the 'new' rating list. Datasets obtained at later dates will continue to be valued at the antecedent date (although individual properties may be revalued at other times on request, or if circumstances change—for example, if the property changes use or is extended). When market value increases or decreases, then the static RVs become out of line with prevailing market rents. This fixed census date also prevents the opportunity to use the dataset to investigate gradual, market-based change in RV from one year to the next.

### 2.3.5. Data Inaccuracies

The research encountered errors in the National Rating List dataset mainly due to inconsistencies in the categorisation of properties within SCat codes. For example, some supermarkets were categorised as retail warehouses; some shops below 750m$^2$ were mistakenly categorised as "large shops"; similar shops were categorised as "shops" and "food shops" in different areas. The research also discovered that some postcode polygon boundaries do not correspond exactly to the divisions of space found in the built environment. In particular, outside dense residential areas, postcodes often subdivide parcels of land and larger buildings, and sometimes do not include properties with the appropriate postcodes within their boundaries.

### 2.3.6. Zero-Rated Properties

A small number of properties in the ratings list are recorded as having a RV of 0 or 1. This applied to relatively few units (less than 2000) but, if larger properties were involved, there was potential for this to skew the results—for example, by bringing down the average RV/m$^2$, or by bringing down the aggregate RV in a given area.

### 2.3.7. Non-bulk Class and "Other" Properties

The starting point for the valuation of most commercial properties is a calculation of the market rental value per m$^2$ of a given type of property in a given area (the tone of the market). However, there are certain property types whose value is calculated by alternative methods such as profits method or contractor's test (depreciated replacement cost). Therefore, their floorspace is not published within the "summary valuations list". This includes some categories of business which may be particularly significant in certain areas, including public houses and bed and breakfast accommodation.

It is worth noting that the "other" category, containing heterogeneous premises that cannot be regarded as a single class of property, does contain some types of business, such as cafes and restaurants, that are important to the socio-economic vitality of town and city centres. When contemplating trends affecting retail and office property, it is necessary to have regard to the interaction between bulk class properties and some "other" subcategories, including those for which floorspace is not recorded.

### 2.4. Description of Study Area

The City of York, in North Yorkshire in England, has a small, compact but pronounced retail core, with most commercial activity focused within the city's unusually complete mediaeval city walls. York has three prominent out-of-town shopping centres: Monk's Cross/Vangarde, to the East of the city; Clifton Moor, to the north; and the York Designer Retail Outlet, to the south. The city is heavily reliant on tourism, being a popular stopping-off destination for international tourists travelling between London to Edinburgh, and domestic demand from its large rural hinterland.

York City Council's own research on the city's economy listed the following strengths: a highly-skilled population; low unemployment; good digital infrastructure; the city's position on the East Coast Main Line with excellent connections to London and Edinburgh; high employment in industrial biotechnology and agri-food research, insurance and rail; its 7 million visitors per year. Weaknesses included: lack of available land for business expansion, low-productivity and part-time jobs, and a mismatch between skills and jobs; congestion on the Outer Ring Road; and competition from other centres. Retail and wholesale made up 16% of the city's employment but 11% of its Gross Value Added. The number of retail jobs was predicted to grow by 2% in the 10 years from 2016 [39]. The study area included the entire conurbation.

As we note above, the aforementioned hexagonal grid was particularly efficacious with regard to the City of York because the underlying (predominantly medieval) street pattern in the centre of the city, in which streets (colloquially known as Gates) generally run southeast to northwest, or at 90 degrees to these, partly due to the River Ouse running through the city in a northwest to southeast direction.

York City Council has produced a new Local Plan (hereafter referred to as the Plan), which is currently awaiting approval from the Planning Inspectorate that assesses local planning policy against national requirements on behalf of the Government. The Plan defines certain areas within the Council's jurisdiction for various purposes—notably, areas within which development is more tightly controlled than elsewhere for the protection of the natural or built environment, and areas where particular types of development are encouraged, such as retail or other employment [47,48].

The Plan establishes a hierarchy of retail centres, at the top of which is the city centre, much of it classified as "primary shopping area", below this are two suburban areas of Acomb and Haxby, classified as "district retail centres", then clusters of shops, known as "local retail centres". Although the Plan acknowledges the existence of out-of-town retail centres at Monk's Cross, Clifton Moor and the York Designer Retail Outlet, it does not allocate their sites for retail, or for any other type of development. Rather, it states that further proposals for out-of-town retail will only be permitted if they cannot be accommodated upon a sequentially preferable site (that is, a site further up the retail hierarchy).

The Plan allocates eleven sites for employment, of which five are classified as "strategic" because they are over 5 hectares in size, despite covering a combined area that is smaller than existing employment areas, as indicated in Figure 2. The Plan implies that established sites should continue to be used for industrial or other commercial purposes but does not explicitly identify which sites are covered by the policy [47,48].

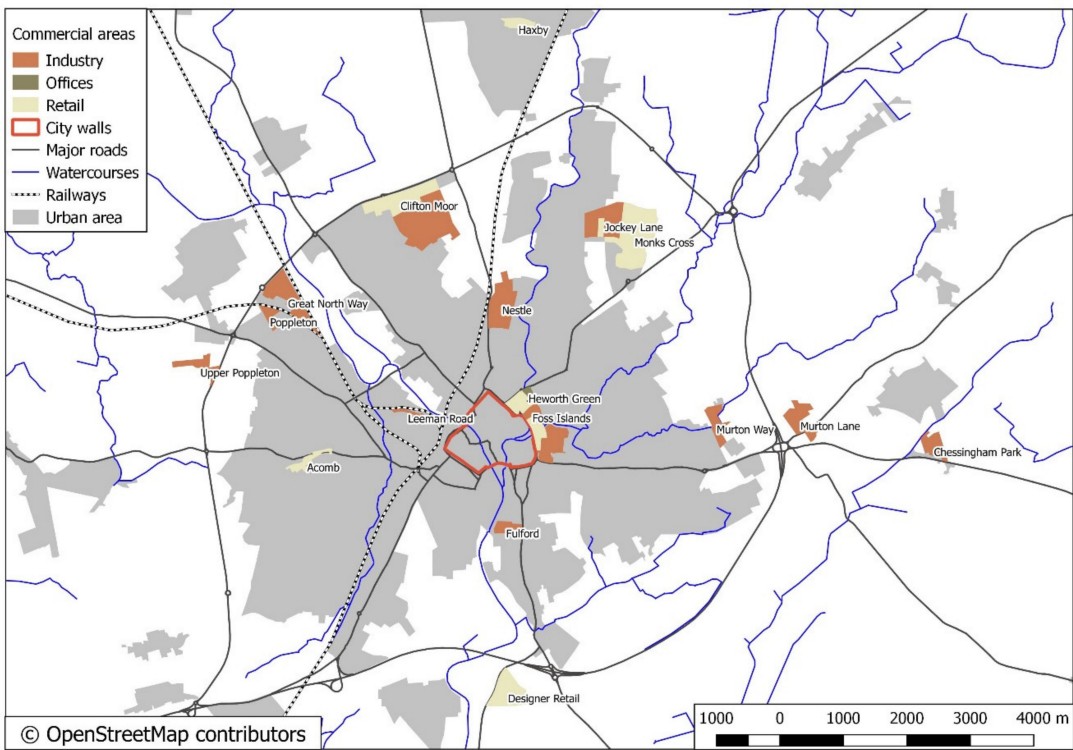

**Figure 2.** Urban footprint, key transport infrastructure and concentrations of employment property in City of York.

Having regard to both the potential and limitations of the data, the following section presents results of the aggregation and spatio-temporal analysis of the stock and value for the three bulk classes of property in the City of York, between 2010 and 2017.

## 3. Results

### 3.1. Variation between "Bulk Class" Sectors

RVs are determined by the VOA on the basis of a property's estimated open market rental value. For "bulk class" properties, this is calculated on the basis of a rental value £ per $m^2$ of floorspace, through comparison with market evidence of the rental values of other comparable properties in the local area—this is referred to as 'tone of the list' which is defined by the Valuation Office Tribunal as:

> *'The general level of value established where a number of properties, similar in size, character, accommodation, quality and location have ratable value have not been challenged or changed or when set on appeal by the Tribunal.'* (paraphrasing Valuation Office Tribunal [49])

There are considerable differences between sectors as shown in Table 1. Predominantly, offices are more valuable than industrial properties, and retail properties are more valuable still, although due to the growth in online shopping or e-tailing in the UK, this traditional hierarchy is under threat. Retail properties also exhibit the greatest range in values (average and standard deviation are large due to the existence of a few small but extremely valuable retail properties in key "prime pitch" locations). The value range of industrial and office properties is less, despite the diversity of the former ("industry" covers everything from storage and warehousing and mineral-producing hereditaments, to computer centres and distilleries).

**Table 1.** RV per $m^2$: York, 2019.

| Sector | Industry | Offices | Retail |
|:---:|:---:|:---:|:---:|
| Average | £53.49 | £118.68 | £278.87 |
| Max | £139.35 | £233.87 | £12,000.00 |
| Median | £52.17 | £117.62 | £187.94 |
| Min | £1.61 | £33.13 | £17.45 |
| Stand.Dev. | £21.41 | £26.17 | £452.34 |

In York in 2019, there was a total of 1.87 million $m^2$ of "bulk class" floorspace, of which 40.5% was industrial, 16.7% offices, 27.5% retail and 15.2% "other". It should be noted that the proportion of industrial floorspace is much lower than it is at a national level. The three bulk classes are now analysed in turn, in terms of distribution of stock, relative value and changes in both distribution and value between the last two rating revaluations.

### 3.1.1. Retail

The study confirmed that RVs for retail properties within the City of York follow a discernible and predictable pattern as shown in Figure 3. The city centre contains a cluster of locations with a large quantum of retail floorspace, with relatively high RV. This is surrounded by zones with intermediate and lower values, as the quality of the retail pitch deteriorates and footfall declines. To the west of the city centre, a low value cluster in the district retail centre of Acomb is apparent. On the peri-urban fringe, two large concentrations of relatively high value retail floorspace to the north and east of the city, can be identified as Clifton Moor and Monks' Cross/Vangarde respectively. These 'out of town' retail parks comprise a relatively small number of large retail units. The RV/$m^2$ in these locations tend to be lower than city centre retail premises, due to a combination of lower land values and larger formats that attract a quantum discount. The isolated York Designer Outlet to the south of the city (indicated by the dark blue column in Figure 3) has different characteristics as it comprises a cluster of small purpose-built units with high RV/$m^2$, which together with a few extremely high value small kiosks increases the average RV/$m^2$. The phenomenon of large out-of-town or edge-of-town retail centres, where higher values prevail due to oligopolistic trading conditions, high convenience and

low costs for users of private motor vehicles is familiar to many towns and cities across the world. The method/approach used to display the data clearly portrays the aggregate size and value of premises located in such centres, relative to the stock and value of 'traditional' city centre retail premises.

The average RV/m$^2$ for retail properties in York, based on the 2017 rating list, is approximately £279 psm. However, this masks a very wide range, from less than £20 psm to over £900 psm for 'prime pitch' units in the city centre, railway station (where there is a captive market) and kiosks in out-of-town retail parks, all of which have high footfall. Predictably, retail floorspace in concentrated in the city centre but also extends along major trunk roads radiating out from the centre. There are several significant clusters located along radial 'trunk' roads, or roads that run through suburban residential areas, indicating a close relationship between retail activity, accessibility and 'local' residential neighbourhoods. The exception to this is York Designer Retail Outlet, which is entirely separate from the city's urban footprint and functions as a premium 'niche' destination that generates its own footfall, almost all of which is car-borne.

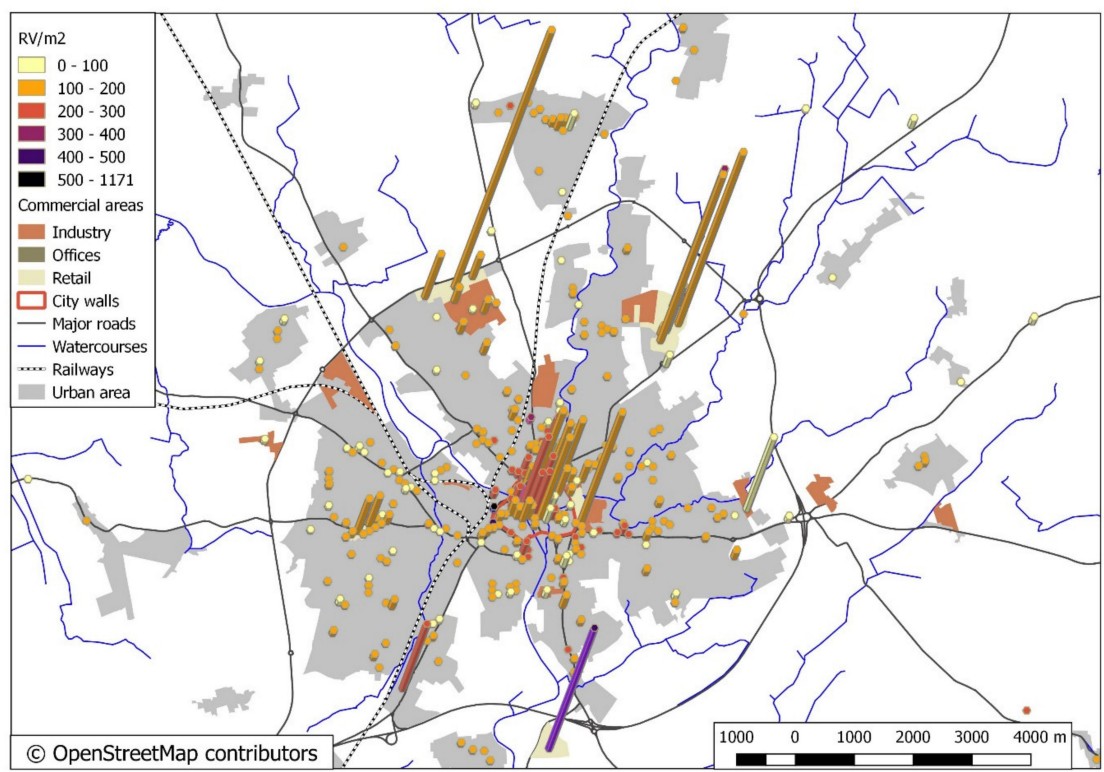

**Figure 3.** Retail RV/m$^2$ (2019 values) aggregated within a 100m grid; extruded to indicate total floorspace.

### 3.1.2. Offices

Office floorspace exhibits a similar pattern of distribution to that of retail, due to both being commercial (rather than industrial) activities, with premises clustered in the city centre, along radial routes, and in peripheral locations in the form of business parks as shown in Figure 4. Again, locations where there is a greater quantity of office floorspace tend to be the same locations where values are higher. However, the pattern of distribution and value of office floorspace is less polarised than that of retail, with a much narrower range of values and more modest spikes in value for premium (or prime Grade A) office locations. The distribution of office floorspace within the city centre is more diffused, with lower incidence of office premises in the historic core of the city. One reason for this could be the limited potential for construction of office buildings with large open plan office floorplates due to the lack of sufficiently large development sites and planning restrictions to protect the sensitive heritage environment (for example, Conservation Area Status). Another important difference is that edge-of-conurbation locations contain a nominally and proportionately lower quantity of office

floorspace than retail floorspace, indicating that the service sector represents a less significant part of York's economy than in towns and cities with larger agglomerations of service sector firms.

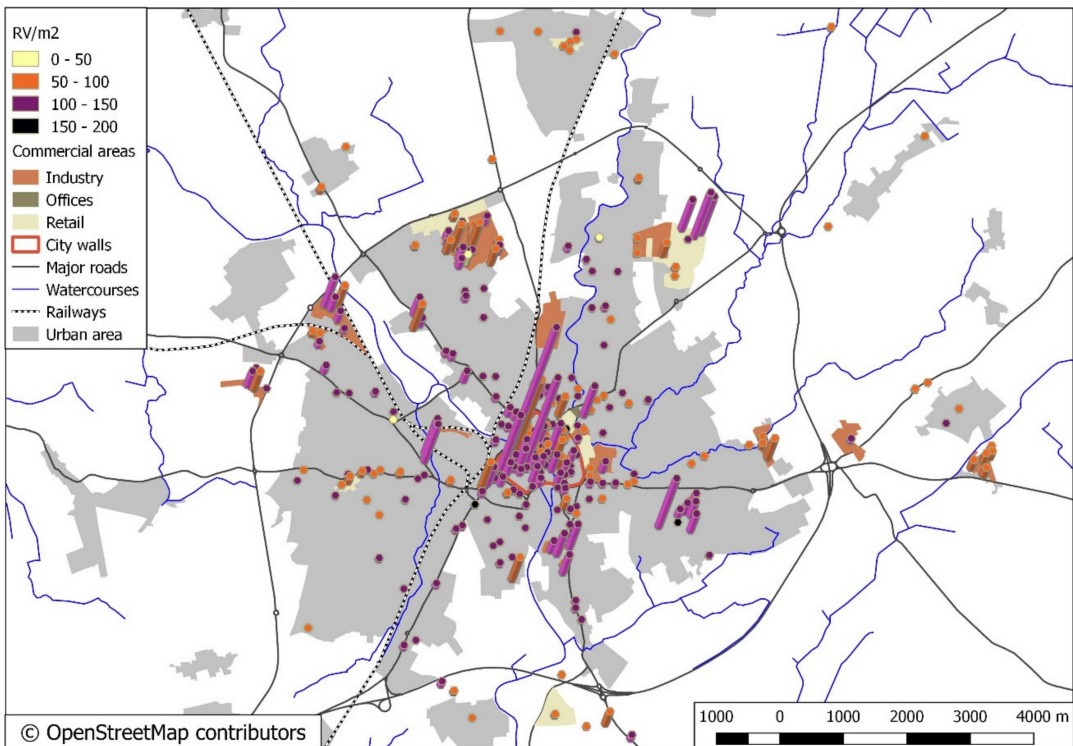

**Figure 4.** Office RV/m$^2$ (2019 values) aggregated within a 100m grid; extruded to indicate total floorspace.

### 3.1.3. Industrial

The distribution and RV/m$^2$ of industrial floorspace in York is more diffuse than for retail or office premises as shown in Figure 5. The two largest concentrations of industrial floorspace are the Nestle chocolate factory, just to the north of the city centre, and the Portakabin depot, to the northeast, adjacent to Monks' Cross. The location of the former reflects York's long association with chocolate making, with the original Rowntrees Chocolate factory being established on the site in the late 19th century, and production continuing following take over by the Nestle Group in 1988. The location of the latter demonstrates the importance, in the 'space hungry' industrial sector, of relatively cheap land in close proximity to major highways. Elsewhere, industrial premises tend to cluster in specific areas such as peri-urban industrial estates at Clifton Moor and York Business Park, and edge-of-centre/zone of transition areas of Foss Islands and Holgate. As noted above, geolocation is sometimes approximate rather than accurate, and this is certainly the case for the Nestle factory. Postcodes are delineated according to residential population rather than area and because industrial properties tend to be found in larger postcodes with lower population density, on the periphery of urban area, aggregation by postcode has the potential to exaggerate their size.

The relationship between the quantity of available space and its RV/m$^2$ is less pronounced than for retail and office bulk classes, revealing that industrial floorspace does not tend to be organised in clusters where both the quantity of floorspace and its RV/m$^2$ are high. This is in part to the incidence of quantum, where the more floorspace that is occupied, the lower the rental value per unit area (effectively the same principle as 'buying in bulk'). Industrial floorspace values are nominally lower and consequently have a much narrower range that either retail or office markets, indicating that 'prime-pitch' or central locations are not important for industrial or manufacturing use.

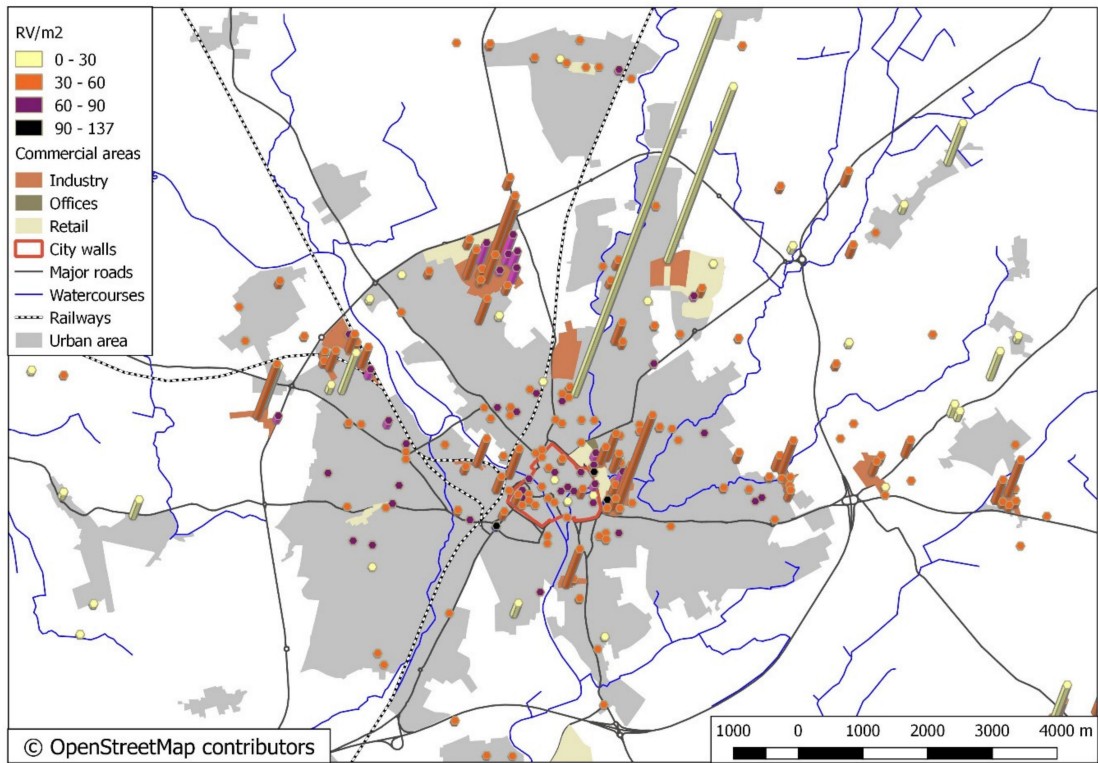

**Figure 5.** Industrial RV/m$^2$ (2019 values) aggregated within a 100m grid; extruded to indicate total floorspace.

### 3.2. Change over Time

One of the advantages of exploiting NNDR data is that it provides a comprehensive longitudinal dataset, as not only is the whole rating list reviewed at periodic revaluations, but the summary list is updated annually as incremental changes are recorded. This creates an opportunity to conduct comparative analysis of the quantity and distribution of floorspace and its relative value between census points, in this case 2010, 2017 and 2019.

### 3.2.1. General Comments

Figure 6 reveals that aggregate retail floorspace increased by 6% between 2010 and 2017; since when, there has been a modest decline. Office floorspace in York declined across both periods, which is broadly consistent with the national picture. The biggest discrepancy with the national trend was in respect to industrial floorspace, which declined by 11% between 2010 and 2017 (approximately 89,000m$^2$) due to the former Terry's Chocolate factory coming off the list, before stabilising between 2017 and 2019 compared to an increase across both time periods at national level.

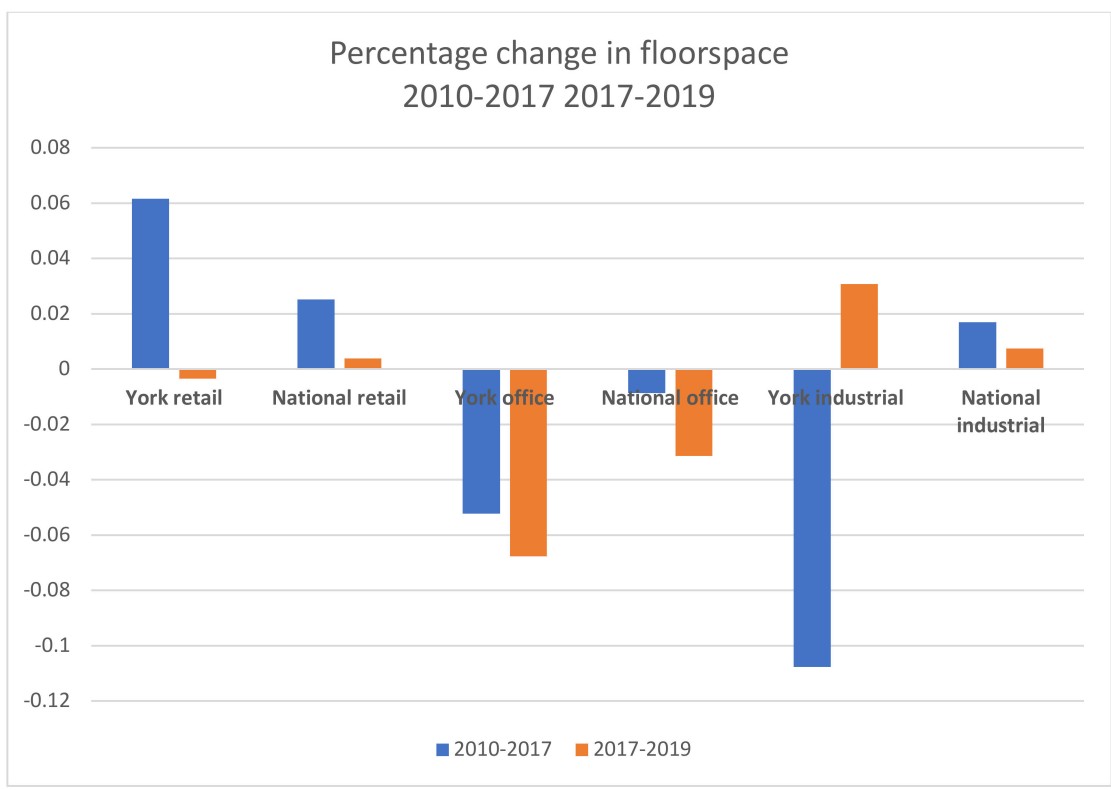

**Figure 6.** Comparative percentage change in aggregate floorspace, York and England and Wales (National) 2010–2019.

### 3.2.2. Retail Property

The modest increase in aggregate retail floorspace in York masks a shift in its distribution that can be revealed by comparing data for each individual grid tile between each census date and where changes occur. Between 2010 and 2017, the condition was one of stasis; of 356 grid tiles containing retail premises, 183 saw no change, 90 saw a decline of less than 500m$^2$, and 57 saw an increase of less than 500m$^2$. The grid tiles in which greater increases or declines occurred exhibit a clear pattern—11 of 13 grid tiles that recorded declines in retail floorspace of greater than 500 m$^2$ were clustered in the city centre; of 13 grid tiles recording increases in excess of 500m$^2$, 2 were in rural locations outside the city, 7 were in peripheral locations (including 5 in out-of-town retail parks), 1 was in a suburb, 1 in an edge of centre retail park. Only two were in the city centre itself. This observed trend for retail floorspace to decline in the centre and increase in the periphery of the city, shown in Figure 7, is often referred to as hollowing out [50]. This trend has continued between 2017 and 2019, albeit to a lesser extent, with all four grid tiles recording increases in floorspace of more than 500m$^2$ being in peripheral locations, two of which were in out-of-town retail parks. Of the five grid tiles registering decreases in floorspace of over 500m$^2$, two were in the city centre, two were at retail parks and one was peripheral as shown in Figure 8.

This is in accordance with the national picture. Across the UK as a whole, most local authorities saw an increase in retail floorspace between 2010 and 2017. Between 2017 and 2019, however, retail growth was generally weaker, with few areas recording increases, and floorspace decline more widespread. Larger increases tended to be in locations where large new retail development had occurred, rather than incrementally through small developments.

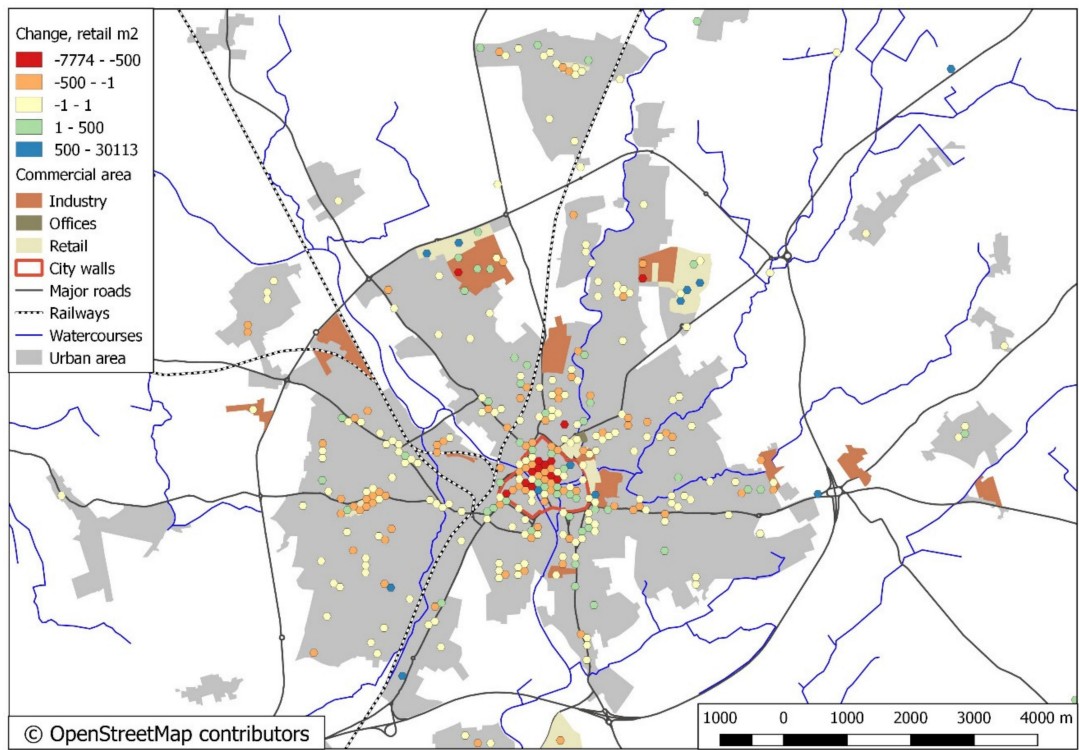

**Figure 7.** Change in retail property floorspace distribution, 2010–2017.

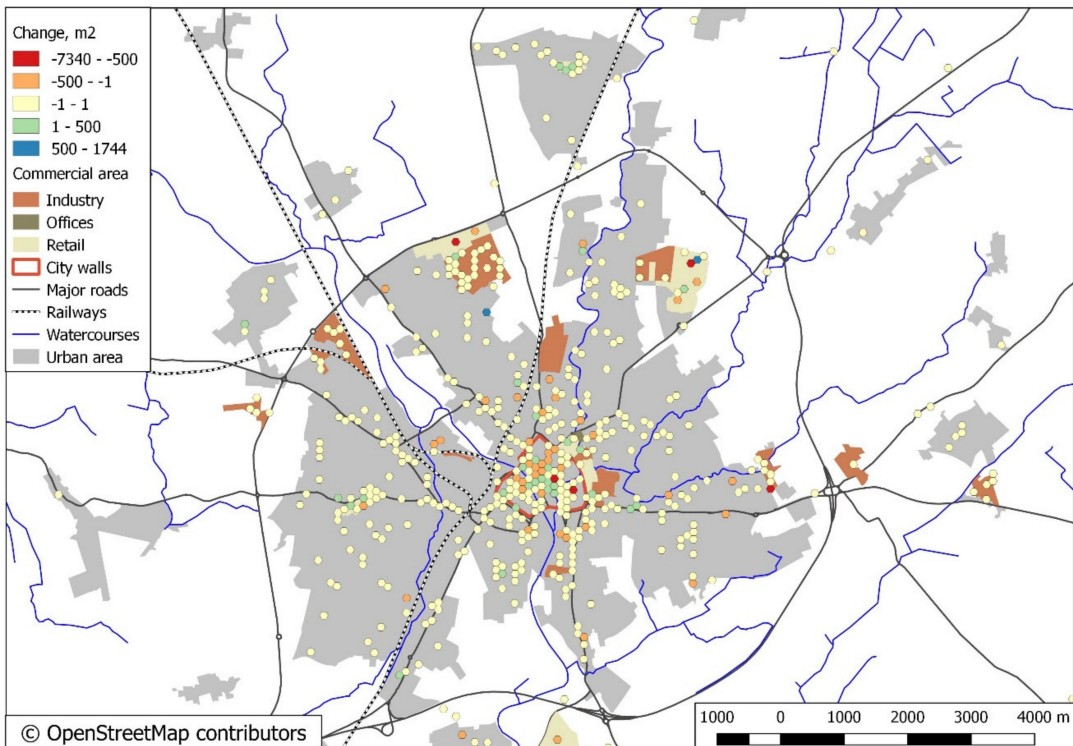

**Figure 8.** Change in retail property floorspace distribution, 2017–2019.

### 3.2.3. Office Property

Office property is the only one of the three sectors that presented a consistent decline in floorspace between the three census points. One potential reason for this might be a change in permitted development rights: since 2013, it has been permissible to change a building's use from office to

residential without planning permission. Where land values are high and residential properties in short supply, this is likely to stimulate a decrease in offices. Here, though, the pattern of variable decline displayed by retail property in Figure 9, does not apply to office property: as the temporal comparison reveals, office premises became more centralised between 2010 and 2017, with four grid tiles located at Clifton Moor recording decreases of more than 500m$^2$. Whilst between 2010 and 2017, there was no apparent growth in peri-central locations, or movement of office floorspace out of the historic core, between 2017 and 2019, the data reveal a tendency for office property to move outside the historic core (within the boundary of the medieval city walls) as shown in Figure 10.

The decline in office floorspace between 2010 and 2019, which was seen at a national level as well as in York, was greater in London and the Southeast, perhaps because this is an area where residential values are high. The general decline in office floorspaces did not, however, prevent some extremely large percentage increases in particular locations.

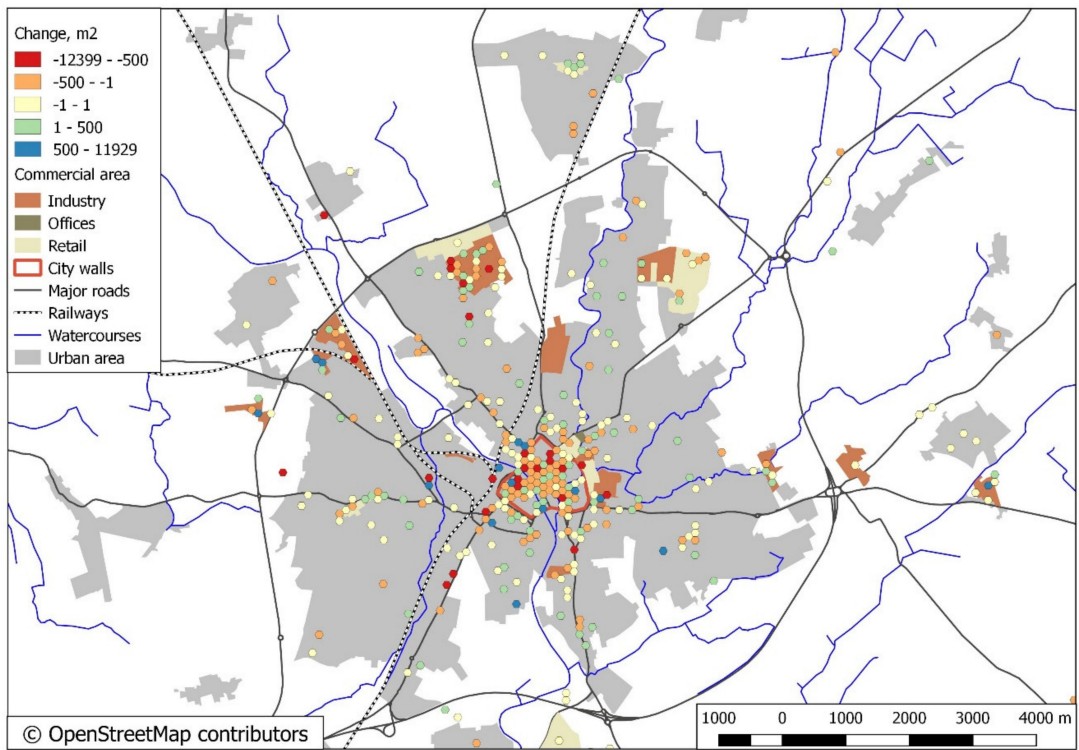

**Figure 9.** Change in office property floorspace distribution, 2010–2017.

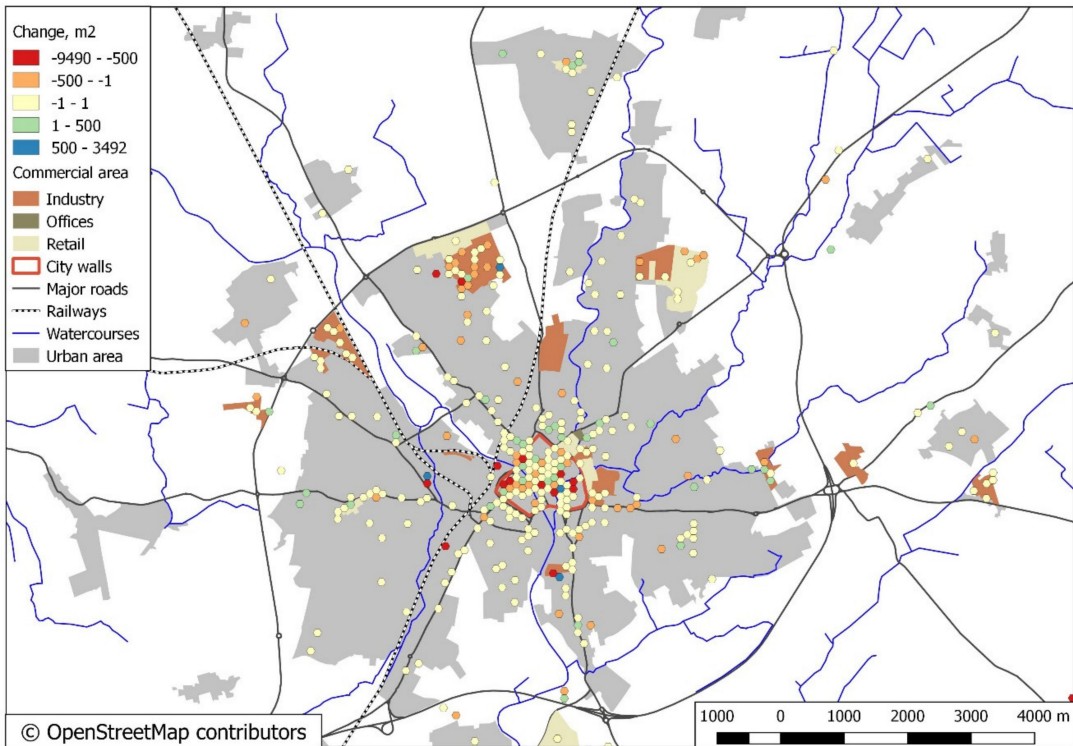

**Figure 10.** Change in office property floorspace distribution, 2017–2019.

### 3.2.4. Industrial Property

The index reveals little if any pattern to changes in industrial floorspace across the three census points. Again, as shown in Figure 11, the general picture is one of stasis: of 327 grid tiles containing industrial floorspace, almost half (154) saw no change. Growth again tended to be focused in peripheral locations and upon established industrial sites: of 11 grid tiles with growth above 1500 $m^2$ (note higher threshold due to industrial premises being larger), 4 were in rural locations, 3 were at Clifton Moor, and 4 at other industrial sites within the urban area. There was no discernible pattern of decline in floorspace, with grid tiles recording decrease greater than 1500 $m^2$ being scattered across the urban area. Whilst industrial floorspace declined by 11%, between 2010 and 2017, two-thirds of which were accounted for by the loss of a single factory, the Kraft Jacobs Suchard complex (formerly Terry's Chocolate Factory) on Bishopthorpe Road, which is being redeveloped mainly for residential use. What little growth in industrial floorspace that did occur between 2017 and 2019 tended to be in rural or peripheral locations. There was no obvious pattern of decline during this period as shown in Figure 12.

Nationally, there was a slight growth in industrial floorspace across the country between 2010 and 2019. Declines were seen in London, perhaps because high land and property values stimulated a change from relatively low-value industrial uses to higher-value ones.

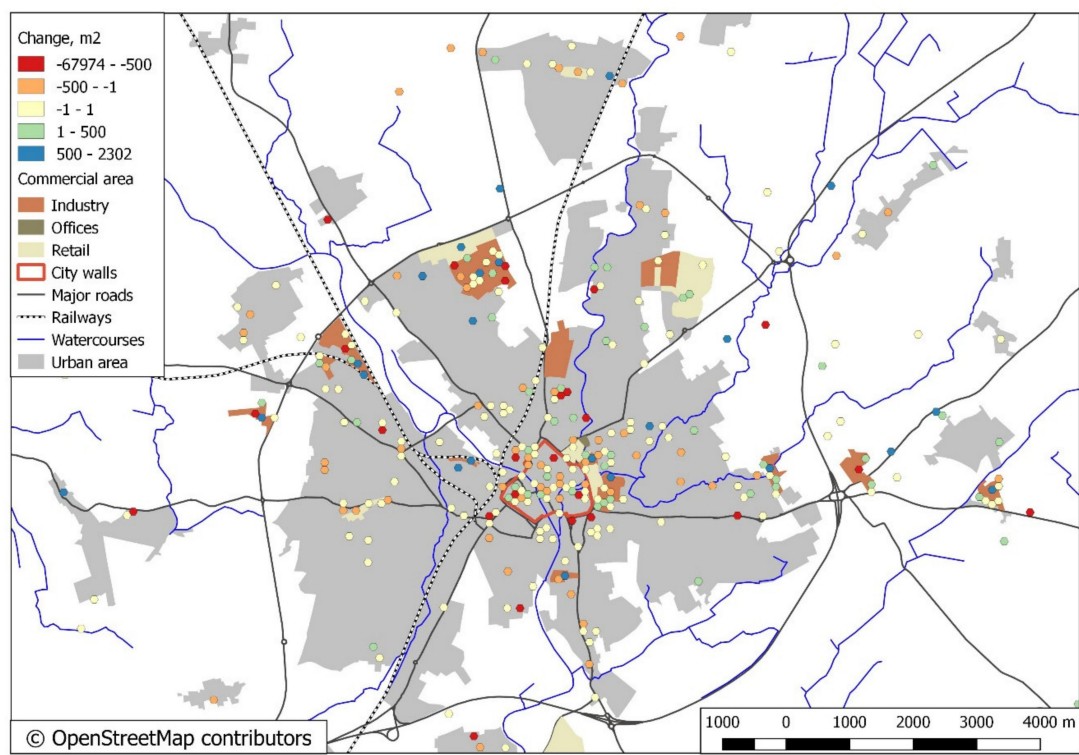

**Figure 11.** Change in industrial property floorspace distribution, 2010–2017.

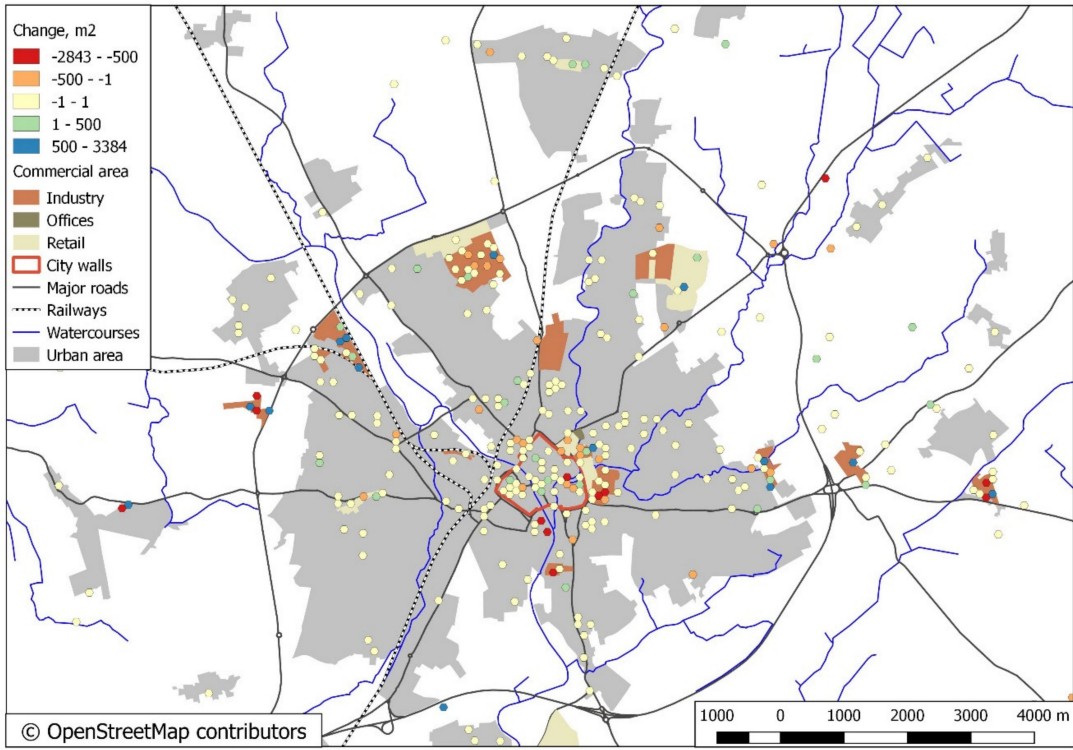

**Figure 12.** Change in industrial property floorspace distribution, 2017–2019.

It may be hypothesised that, as retail and office occupancy declined in the city centre, the space vacated might be filled by other uses. Uses such as public houses, cafes and restaurants, clubs and theatres are described as "main town centre uses" within the National Planning Policy Framework [51], and might be regarded as having some locational characteristics in common with retail activities.

In order to investigate this hypothesis, records were grouped together in SCAT codes, pertaining to food and drink, to investigate patterns of change. Evidence tends to support this hypothesis, with an increase in the number of food and drink premises across the city as shown in Figures 13 and 14: 418 in 2010, 454 in 2017, 475 in 2019. There were 219 grid tiles which contained any food and drink premises between 2010 and 2019, of which 61 were wholly or partially in the historic city. In total, 15 grid tiles saw large increases (over £100,000) in the value of food and drink premises between 2010 and 2017, of which were in the historic city and 4 out of 5 between 2017 and 2019. The exceptions were new food and drink premises at Vangarde and Monk's Cross out-of-town shopping centres.

In fact, the total RV of food and drink property within the city walls saw a considerable increase between 2010 and 2017, and continued to rise thereafter. The increase in food and drink significantly compensates for the loss in retail RV, but it does not compensate for all commercial property losses in the city centre between 2010 and 2019. Daytime patronage by service sector employees is important in sustaining food and drink retailers in a town or city centre. The growth in food and drink retail in York has been fuelled by its strong tourism industry, however, decline in office space over the same period could be a source of vulnerability, especially if tourist numbers were to decline.

There was a very large increase in the RV of non-food and drink commercial property outside the city centre between 2010 and 2017—£12.8 m—even though this category included office property, which saw a decline of £3.2 m. The increase was made up of the following: £8m of retail property; £3.6 m at the expanded University of York; £2.1m of new hotels (across the city as a whole, hotel RV increased by 48%); and smaller additions to the National Railway Museum (£0.9 m) and York District Hospital (£0.5 m).

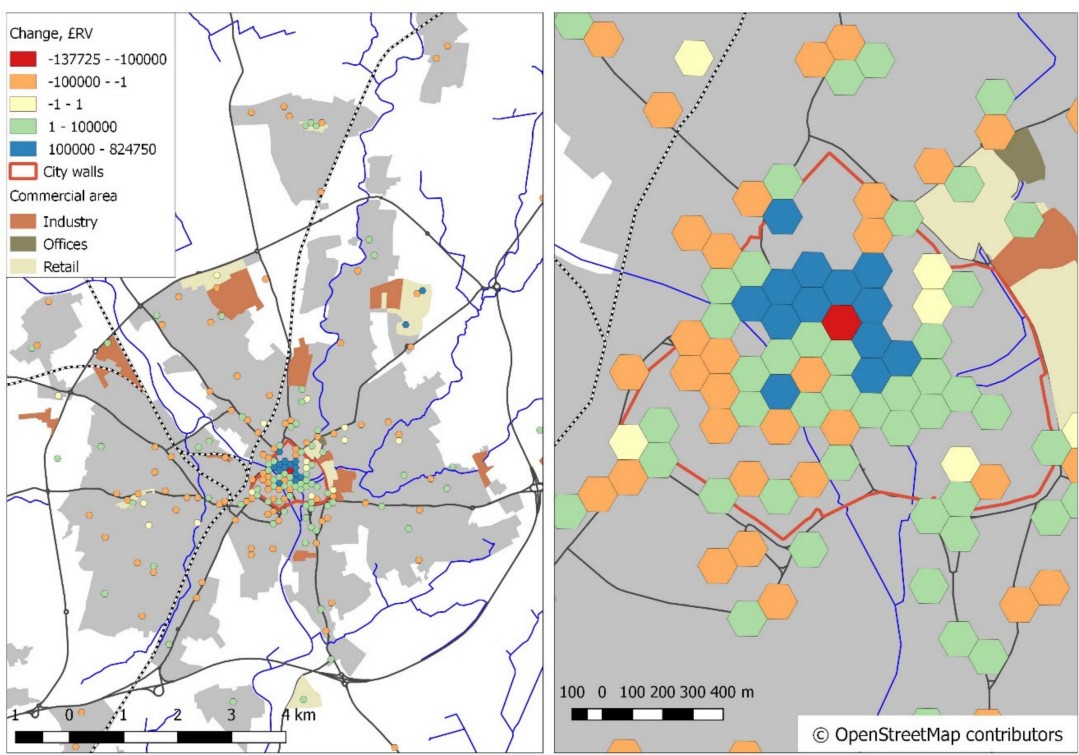

**Figure 13.** Change in food and drink RV, 2010–2017.

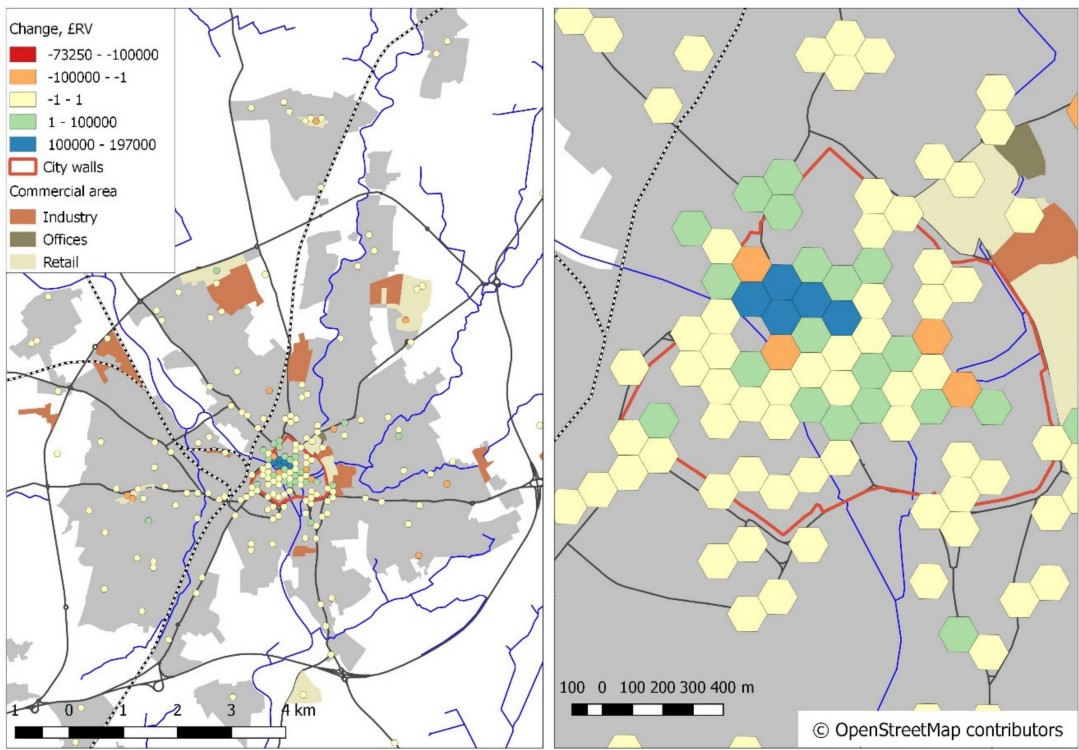

**Figure 14.** Change in food and drink RV, 2017–2019.

## 4. Discussion

The study has demonstrated the potential opportunities for national and local governments to exploit national non-domestic rating (NNDR) data to capture and represent changes in the quantum, spatial distribution and relative value of employment floorspace by sector e.g., retail, office and industrial premises over time. The use of this comprehensive dataset means that all premises, regardless of size, status or sector, are captured. The authors propose that RV/m2 can be used as a proxy for market attractiveness, since it provides a consistent measurement of the value of properties of different sizes in different locations.

The study revealed that whilst retail and office agglomeration persisted in the city centre, consistent with classical commercial property location theory [20,22,23], there were significant clusters of commercial floorspace on the peri-urban fringe, particularly at intersections between radial transport corridors and the city's ring road. The result of this effect is an emerging binary commercial property market generating increased vehicle movements, not only by customers but also employees working in retail and office sectors. The industrial sector in York, by contrast, is relatively small, dominated by two large concentrations of manufacturing floorspace, and has recorded a decline in floorspace that is inconsistent with the national trend. Of all property sectors, location is most critical to retail property, relying as it does on footfall and pedestrian flows; it is also the sector that appears most vulnerable to change. Temporal analysis of rating list data has revealed that polarised growth of out-of-town retail has created a binary commercial property market that risks hollowing out the historic city core. Further refinement of the methodology allowed data to be segmented within bulk class categories, for example retail bulk class data could be segmented between traditional retailing and the food and drink sector. Analysis confirms that York's city centre is changing, with increased representation of food and drink uses that, to some degree, has compensated for the hollowing out of traditional retailing. The methodology also offers potential to differentiate between smaller and larger shops, convenience and comparison shops, or higher and lower order goods.

The methodology presented in this article could usefully be employed in the administration of land and property by a variety of stakeholders, including central and local government, regional, sub-regional and metropolitan authorities, economic development agencies, spatial planners, economic development strategists and town and city centre managers. It may also prove useful to real estate investors, landlords and the firms and businesses that occupy employment floorspace. The method creates opportunity to explore relationships between the performance of employment floorspace and wider socio-economic factors such as technological change, government fiscal and spatial policy, transport and demographics, all of which have an impact on the future viability of commercial and industrial property markets.

Geospatial data have an important role to play in understanding prevailing physical, economic and social conditions within urban areas and offer a way of exploring and representing the interrelationship between the location of fixed infrastructure and built environment (physical) assets, and more dynamic patterns of market performance (economic) and movements (social). The use of a large corpus of up-to-date data can improve our ability to model the potential outcomes of anticipated urban interventions, perhaps as part of scenario modelling or options appraisal. As Thompson et al. [52] observe, urban policy makers and planners are hampered by a mismatch between the future world in which their decisions will take effect, and the past world on which they have information:

> "*planning (for the future) is always conducted through the rear view mirror.*". (Thompson et al. (2016) p. 81)

The case study presented in this article demonstrates that, whilst this might be true, there is big data available in the UK with which to accurately track changes in the distribution and relative value of prevailing land uses over time and space. Having comprehensive and accurate quantification of the stock and value of all employment floorspace would inform local government officers, surveyors, property managers and planners when conducting land administration, spatial planning, economic development, and other policy making. We have shown that geospatial analysis is a powerful tool that can 'add value' to existing big datasets, by mapping and analysing the quantum, spatial distribution and relative value of commercial and industrial floorspace across an urban area relative to other spatial characteristics. The analysis of rateable value and floorspace data also affords the opportunity to investigate the relationship between economic performance and variables for which spatial data are available such as transport networks, flood risk, urban form and population distribution. The use of temporal data across census points offers the opportunity to map and analyse patterns of change, revealing which sectors of the economy are growing, in terms of the quantity of floorspace occupied, and which are declining. Analysis employing such data can be used to test whether 'traditional' theories of industrial and commercial location are still as relevant in economic sectors that are prone to disruption through technological advances and mobile communications. The next steps are to exploit NNDR data to replicate the methodology for other towns and cities in the UK to permit comparative analysis of the performance of different locations, over time, to determine whether trends in the quantum, distribution and value of employment floorspace observed in York are being experienced more widely.

**Author Contributions:** Project lead, Paul Greenhalgh was principal investigator, lead author and provided input on commercial property markets, planning and development, valuation and rating aspects of the research, also responsible for writing, reviewing and editing the article. Method development, Helen King, is a GIS specialist and provided particular input around method development, analysis of data and illustration of results. Business rates expertise, Kevin Muldoon-Smith is a specialist in UK non-domestic rating and policy reform who helped refine the cleansing, matching and enhancing of national non-domestic rating data. Case study, Adejimi Adebayo contributed to the identification and preparation of the York case study including data extraction, removing duplicate records and mapping of spatio-temporal change in variables. Josephine Ellis was principal GIS developer and responsible for generating images, analyses and result from GIS and preparing the original draft of the article. All authors have read and agreed to the published version of the manuscript.

**Funding:** This research received no external funding.

**Acknowledgments:** We would like to acknowledge the contributions of Obinna Anejionu and Bradley Sparkes to early GIS method development and Gavin Chait of Whythawke for advice on dealing with duplicate records.

**Conflicts of Interest:** The authors declare no conflict of interest.

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
