# Peer review of "Using GIS to Explore the Potential of Business Rating Data to Analyse Stock and Value Change for Land Administration: A Case Study of York"

_ijgi, doi:10.3390/ijgi9050321_

Round 1

Reviewer 1 Report

It is an interesting article, but I have some comments. "GIS" is today a range of systems for visualisation, analysis and presentation of location based data, which can be almost everything.

I think the use of "GIS" does not say so much and you may improve the article by specifying more in detail what you are doing. 

The introduction is rather "wordy" and may benefit from being shortened somewhat.

Author Response

The authors have reviewed our use of the term GIS and have further elaborated on the context for the use of GIS for land administration and our explanation of the use GIS to visualise and analyse business rates data in both the introduction and method sections.

With regard the length of the introduction, the authors felt that it is necessary to introduce, in sufficient detail, the nature of the domain within which GIS was being used in a wider land administration context. In particular, with regard the specific characteristics of the dataset that was being exploited, as this is important in understanding both the potential and limitations of what can be achieved in terms of analysis and visualisation.

Reviewer 2 Report

Row 63 - parentheses after the citation 5 is superfluous.

Row 72: please define terms ODPM and CASA with full name and then you can use abbreviation.

Row 289 and 290 - please define term RV

In whole text some measure units (for example meters) need to express with one space (for example row 344 - please fix that in whole paper where ii necessary).

Figure 3 is not a figure. It is a table. Please fix that and take care for other figures (changing numbers of figures).

Do not use term "See figure 3 bellow". Follow the MDPI IJGI instructions for authors.

I suggest in Sectio 3.2. add two sentences for explain the purpose of section 3.2.

Author Response

The authors have addressed all the corrections as requested.

Reviewer 3 Report

The authors provided a nice study, the results are also good and acceptable. I have some remarks but not to the content just according to the organizing the text. In my opinion, it is not enough to write good results, it also important to "sell" them. I mean, with iterating the content to the scientific writings standards. 

My general comment is the lack of scientific citations (beside the other ones). It would be very desirable to have more as it makes more convincing that the authors know the results of previous studies.

I suggest to reorganize/rewrite the abstract: it should a short paper with 1-2 sentences of introduction, the small summary of the methodology and the results, finally, a sentence about why are these results good. More or less most required content is there, but the counting is not appropriate, and please consider to rephrase some parts.

In the introduction, it may have sense to provide an outlook to the world, how the economic processes happen in the world. A common mistake to focus on the given area and global description is missing. However, it is a good tool to introduce and cite other researchers’ works who will also read and possibly cite this study, too. On the one hand the paper will be more precise and on the other hand the authors can grab more citations with a wider audience. Now, e.g., the first paragraph there is no citation at all.

The methodology and the results sections are nicely written and provide all important information.

My other note is that the Discussion is not in the right structure. Please follow these suggestions and rewrite this section.

  • please do not write general statements, especially without citations: the discussion is directly for discussing the results
  • from point to point (of course the relevant ones where the is a sense to write about) please explain the causes and consequences of the results
  • all statements are convincing if proved with citations, results of previous studies: this topic is not novel, there should be authors applying different methods in different cities, there can be similarities and differences (maybe controversial results) but it is the real world, all similar or controversial results are welcome if the authors can explain the causes
  • now, this section does not provide the above requirements, please try to use my suggestions

… and the Conclusions section is completely missing. I give some advice according to a possible content:

  • revisit the aims in one sentence
  • enhance the novelties of the approach, and the findings of the results (even using bullet points)
  • a closing sentence for possible future works and users

Author Response

The abstract has been re-written as requested.

Additional (10) citations have been introduced both in the introduction and method sections to identify further studies that have informed the research project.

With regard the content of the discussion section, this has been re-edited to provide more coherent explanation. Due to the edition being a ‘Land Administration Special’ the authors have sought to maintain wider relevance to this particular theme. Please note that the research is highly original and there are no recent examples of the analysis and visualisation of such a (UK) dataset using GIS, with which to compare.  Other studies that have influenced the development of the method and application have been identified and referenced in the introduction and method sections, however there are no directly comparable contemporary studies with which to compare results, which are specific to a UK real estate market context.  

Journal guidance template indicates that a conclusion section is only required when the discussion section is long or complicated, which it is not. The authors, on the advice of the Journal, have therefore deemed that the addition of conclusions section is not necessary.

Round 2

Reviewer 3 Report

Dear Authors,

I appreciate the changes and the efforts to make the maniscript better and I accept it.